# Quantification and Environmental Assessment of Wood Ash from Biomass Power Plants: Case Study of Brittany Region in France

Karine Dufossé, Marine Marie-Charlotte, Vincent Augiseau , Thierry Henrion and Hayet Djelal *

UniLaSalle-Ecole des Métiers de l'Environnement, CYCLANN, Campus de Ker Lann, 35170 Rennes, France; karine.dufosse@unilasalle.fr (K.D.); marine-mc-97122@hotmail.fr (M.M.-C.); v.augiseau@citesource.fr (V.A.); thierry.henrion@unilasalle.fr (T.H.)

\* Correspondence: hayet.djelal@unilasalle.fr

**Abstract:** The increasing demand for energy is leading to the increasing use of renewable resources, such as biomass, resulting in the significant development of the wood energy sector in recent years. On the one hand, and to a certain extent, the sector has generated many benefits. On the other hand, the challenges related to wood ash (WA) management such as increasing tonnages, landfilling, restrictive regulations for reuse, etc., have been weighing more heavily in the debate related to the wood energy sector. However, all studies have assumed that no environmental impacts can be attributed to WA production. This study aims at discussing this assumption, whether the WA is a waste or a co-product of heat generation. In the first place, WA deposits were estimated using the biomass database and ash content from the literature regarding the collective, industrial and tertiary biomass power plants (BPP) in the French region of Brittany. Then, the impacts of the generated WA were estimated using the attributional life cycle assessment (LCA) method through two different impact allocation procedures (IAP), "from cradle to gate" (excluding the waste treatment). In Brittany, for the year 2017, an estimated amount of 2.8 to 8.9 kilotons of WA was generated, and this production should increase to 5 to 15.7 kilotons by 2050. The LCA conducted through this study gave an emission of 38.6 g $CO_{2eq}$/kW h, with a major contribution from the production of the wood chips. Considering the environmental aspect, the IAP analysis indicated that energy and economic allocations were not relevant, and that, using the mass allocation, the environmental production of WA could represent 1.3% of the impacts of the combustion process in BPP. Therefore, WA, and especially the fly ash, can be considered as a waste from BPP heat production, without any environmental impact attributed to its generation.

**Keywords:** biomass power plant; wood ash; Brittany region; LCA allocation; by-product

## 1. Introduction

The biomass burnt in biomass power plants (BPPs) constitutes an important alternative to fossil energies for the production of heat which, in a very few cases, can be combined with the production of power [1]. Wood biomass is considered a $CO_2$-neutral source of energy because, while growing, wood nearly absorbs the amount of $CO_2$ released when burnt [2]. Wood biomass is found in different forms, such as forest wood chips, sawmill residues, end-of-life industrial wood or pellets, which can be burnt. Regrettably, higher amounts of ash are produced when using the raw materials of biomass for the production of energy [3]. Very few data have been collected on ash production and valorization, but it was estimated that 330,000 t–1,000,000 t of ash were generated by French biomass utilization in 2020 [4]. The wood ash composition varies greatly according to the biomass composition, the combustion technology used and other parameters, but industrial wood ash usually contains multiple oxides in various proportions: $SiO_2$, $CaO$, $Al_2O_3$, $Fe_2O_3$, $Na_2O$, $Al_2O_3$, $Fe_2O_3$, $MgO$, and $K_2O$ [5,6]. Wood ash is frequently disposed of as landfill, but it can also be used in forests as a nutrient to compensate mineral loss [7]. The effects of adding

wood ash (WA) to soil have been found to be negligible, both in term of pH and biological changes in lower soil layers [8]. WA is composed of fine particulate matter which can be a potential health risk to nearby residents [1]. Therefore, it is interesting to develop and assess other ways of valorization. Wood ash not only results from the production of energy; it is also a material that produces secondary resources, i.e., resources taken from the anthroposphere. It can partially or totally substitute primary resources extracted from the natural environment. Using secondary resources is often referred to as urban mining, a concept defined as the "systematic exploitation of anthropogenic materials from urbanized areas" [9]. These areas can be seen as "huge, rich and diverse mines of raw materials". Using secondary resources contributes to rebalancing material flows between urban areas and their hinterland. In addition to reducing the pressures on primary resources and the environmental impacts caused by the extraction, transformation and transportation of materials, it also reduces the pressures on the environment and land-use conflicts related to the landfilling of waste. Moreover, it can contribute to circular economy strategies where looping material flows are generally considered as a key action. Limiting some of the environmental impacts of wood energy production can help with transitioning to renewable energy and can also contribute to returning biological resources to the biosphere [10–12].

The rapid growth of urbanization and construction is increasing the demand for cement. However, Portland cement used in mortars and concretes all around the world is one of the most polluting construction materials. About 7% of the global $CO_2$ emission originates from the production of Portland cement—and approximatively 650–920 kg of $CO_2$ is emitted during the production of one ton of cement [13,14]. The construction sector needs alternative and replacement materials for cement, due to its negative effect on the environment. Thanks to its pozzolanic proprieties, biomass wood ash shows good results at replacing a part of the cement used in the formulation of concrete. Pavlikova et al. (2018) observed a decrease in both $CO_2$ and energy consumption when the amount of wood ash is increased in the mortar mix [15].

Since the exponential growth of sustainability research in the industrial sector over recent decades, companies have increasingly resorted to life cycle assessments (LCA) as a tool to quantify the environmental and energy burdens of their products and services [16]. LCA is a standardized method used to measure these impacts through the whole life cycle of the system (product or service), from raw material extraction to disposal, by way of manufacturing and usage stages [17]. The method can be used, for instance, to determine the viability of using alternative resources in industrial processes or to measure the environmental footprint of a product. In the case of processes generating multiple products, allocations can be performed to divide the energy consumption and environmental impacts between the main products and co-products through the determination of an allocation coefficient [18]. This coefficient has a potentially significant impact on the overall LCA outcome.

Studies that assess and compare WA treatment (landfilling, fertilizing or valorization within other industrial sectors) through LCA assume that no environmental impact can be attributed to the production of WA, as it has the status of waste [15,19,20]. As WA can be valorized as a resource within multiple industrial sectors, it could be considered as a co-product of the BPP, with heat as the main product. The present study thus aims at using the results of a BPP LCA to estimate if and how much some environmental burden can be attributed to WA at the door of the plant if considered as a co-product. A methodology is established to first estimate WA deposits in a territory using calculations carried out on the basis of hypotheses related to the chemico-physical properties of wood fuel and ashes. Then, by-product allocations (mass, energy, economic) are performed, compared and combined with the LCA method to assess the environmental burden of the WA. The geographical scope is restricted to Brittany, a French region located in the Northwestern part of France, where the wood energy sector is quite developed and experiencing a continuous and sustained progression. Indeed, since 2005, the reference year of the Regional Air

Climate and Energy Schemes (RACES), the number of BPPs has been multiplied by almost 6 in Brittany, reaching 509 plants in 2018 [21].

## 2. Material and Methods

### 2.1. Brittany Region Case Study Description

Brittany is one of 22 mainland French regions, and the most western one. Its surface area is 34,023 km$^2$ and it counted 3.3 million inhabitants in 2019. As a peninsula, this region is surrounded by the Atlantic Ocean on the South and the English Channel on the North. Due to a succession of funding plans over the past decades, 550 BPPs can be counted in the Brittany region as of the end of 2020, with nominal power ranging from 15 kW to 33 MW. Therefore, the biomass consumption rose to 552 kton for a generation of 394 MWth by the end of 2020. This activity is deeply localised within the territory as it is using local biomass; it created 420 jobs and it allows a reducation in greenhouse gaz (GHG) emissions of an estimated 310 kton of $CO_{2eq}$/year [22].

Most of Brittany's BPPs use wood as fuel. This study only deals with WA from collective, industrial and tertiary BPPs using wood in all its forms (wood chips, pellets, sawmill by-products, end-of-life wood from wrapping, demolition or industries) as fuel [22]. Consequently, collective, industrial and tertiary BPPs using biomass fuels other than wood (miscanthus, corn cobs, etc.) as well as residential heating are not considered. Plants using pyrogasification processes instead of direct combustion are not considered either, due to the novelty of the process on the French market [23]. Multiple sources of biomass are used to feed this BPP industry: forest wood chips (31%), pruning wood (16%), bocage wood (13%), residues from wood industries (10%), end-of-life wood products (27%) and green wastes (3%) [24].

### 2.2. Estimation of Wood Ash Deposits

The conditions in which the combustion process unfolds have a huge influence on the chemico-physical characteristics of WA especially regarding ash rate, repartition per type, composition and also quality [25,26]. Most of the influential factors are inherent to the wood fuel features (type, ash content, moisture content, etc.), the equipment (type, size, etc.) and operating conditions (type of ash collection, temperature of combustion, etc.) [4]. They must be considered in estimations of WA production.

As data related to WA deposits in Brittany are not directly available, estimations have been made to determine both global production and individual production per BPP. The calculations used are based on recommendations of the BPP guide published in the framework of the "Rules of Art Grenelle Environment 2012" French program (Equations (1) and (2)) [27]. Consequently, only four influential factors are directly and indirectly considered to estimate the WA production in both regions, namely the tonnage, the moisture and ash content of the wood fuel consumed, and the type of WA collection (dry or wet). These factors have a strong influence especially on the parameter "quantity" of WA production [4,28].

Dry recovery:

$$T_{wa} = C_{wood} \times (1 - h) \times t_{wa} \tag{1}$$

Wet recovery:

$$T_{wa} = 1.4 \times C_{wood} \times (1 - h) \times t_{wa} \tag{2}$$

With:

- $T_{wa}$: Annual tonnage of WA produced (t·year$^{-1}$)
- $C_{wood}$: Annual tonnage of wood fuel consumed (t·year$^{-1}$)
- h: Average moisture content of wood fuel (%)
- $t_{wa}$: Ash content of wood fuel (%)

Estimates have been made for 2017 as the reference year and adapted to both BPP and regional scale, considering respectively WA tonnage per BPP and regional WA tonnage.

Data related to wood fuel consumption ($C_{wood}$) for both scales were recovered from AILE database in Brittany [24].

In addition to these datasets, other assumptions were made to get as close as possible to the actual situation. Table 1 gathers the main assumptions considered for the calculations of WA tonnages. A grate furnace has been considered for the wood combustion process, this technology representing the main type of biomass boiler used in France as in Brittany [4]. This assumption has a direct impact on the mass distribution of ashes, based on the way of collecting them (80% of bottom ash, 20% of fly ash) [29]. Moreover, several ash content values ($t_{wa}$) were selected according to the type of wood fuel used, varying from 0.5% (low hypothesis) to 3% (high hypothesis) of incoming wood fuel [4,28–30] with an average moisture content (h) of 30% considered appropriate based on data in the literature [31]. In the absence of a detailed inventory of the distribution of dry and wet WA collection in the studied territory, an overall dry ash tonnage was estimated, to avoid overestimation of tonnages. If these data were available in future years, the relevant dried fraction of WA would have to be multiplied by a factor of 1.4, as presented in Equation (2).

**Table 1.** Main assumptions considered in the estimation of annual wood ash production in Brittany.

|  | Low Hypothesis (LH) | High Hypothesis (HH) |
| --- | --- | --- |
| **Wood fuel consumption per type (Brittany 2017/2050)** | Forest and bocage wood chips: 63% (annual consumption 162.440 tons) Sawmill waste: 10% (52.400 tons) Wood waste: 27% (141.480 tons) [23] | |
| **Ash content per wood fuel type ($t_{wa}$)** | Forest and bocage wood chips: 1.5% Sawmill waste: 0.5% Wood waste: 0.5% | Forest and bocage wood chips: 2% Sawmill waste: 1% Wood waste: 2% |
| **Combustion technology** | Grate furnace | |
| **Type of WA collection** | Dry collection | |
| **Wood fuel moisture content (h)** | 30% | |

Beyond the estimation of WA deposits for the reference year, the analysis also studies the development prospects of the wood energy sector in Brittany which are commonly defined by the Regional Air Climate and Energy Schemes (RACES) for the 2050 horizon [21]. If the consumption of wood fuel increases from 2017 to 2050, it is, however, assumed that the distribution of wood fuel consumed per type does not change.

According to CEDEN (2019) [32], for grate furnace combustion technologies, the produced ashes are split between bottom ash (80%), coarse ash (15%) and fly ash (5%). Other combustion technologies (spreader stoker and fluidized bed-boiler) do not produce coarse ash but a majority of fly ash (60% and 90% respectively), with less bottom ash (40% and 10% respectively).

*2.3. Assessment of the Environmental Burden of Wood Ash*

2.3.1. Life Cycle Assessment and Impacts Allocation Procedures

This study presents an LCA of a complete BPP in Brittany that is producing energy as well as WA. The aim is to assess the environmental burden that could be attributed to WA within the BPP, as a basis to, in further studies, estimating the impact reduction of the replacement of some cement by WA within a concrete mix. WA is thus treated as a co-product in this evaluation, not a waste flow. The European standard EN 15978:2011 [33] only refers to Attributional LCA for the construction sector. The boundary of the studied system includes only the BPP and its supply (Figure 1), but excludes the waste treatment. This is thus a "cradle to gate" study to evaluate the potential environmental impacts of

producing heat and WA. The defined functional unit was thus "to produce 254.7 MWh of heat and 1 ton of wood ash with a 5MW biomass power plant in Brittany region, France".

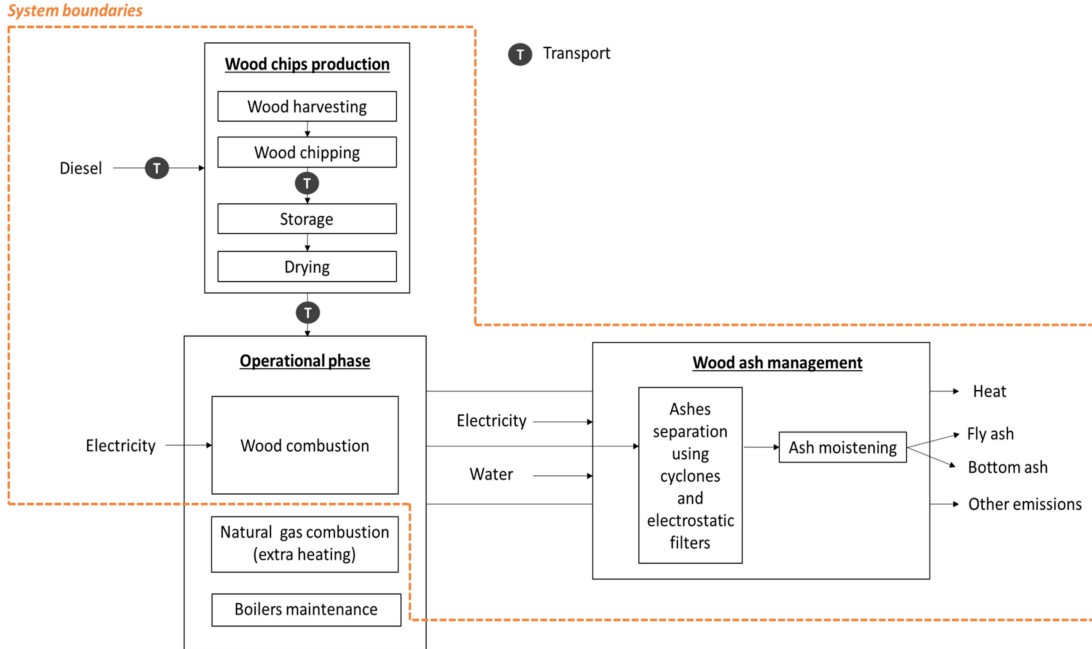

**Figure 1.** Flowchart of the considered system boundaries. Note that dark circles with mention "T" refer to transport.

LCA standards suggest using allocation as a last resort in order to obtain the most reliable results [16]. Indeed, allocation can be avoided by splitting processes into more detailed sub-processes. When this cannot be done, a system expansion method is the most desirable to allocate the environmental burden of products and co-products. This method leads to an expansion of the core system to include alternative ways of producing co-products. Indeed, the aim of system expansion is to assess "consequences of a change in demand" by a consequential approach of LCA which considers socio-economic aspects that are difficult to quantify properly and so potentially generate high uncertainties [34]. Therefore, this work has been focused on Attributional LCA and co-product allocation.

Compared to the economic allocation, the physical allocation procedures (mass, energy) are relatively constant over time due to low changes of mass/energy ratio between product and by-product(s) except in cases of technical innovation. However, most studies consider the economic allocation procedure due to "its simplicity and its ability to illustrate the properties of complex systems" [35], with other approaches used as a last resort [16]. This method has the disadvantage of instability because of market price fluctuations. It is thus interesting to compare and discuss the results obtained through these different allocations methods.

LCA and impacts allocation procedures (IAPs), two widespread methods, have been combined to estimate the environmental burden of WA. LCA was first used to model the wood combustion process, considering all related inputs and outputs but also assessing its environmental impacts. It is today a very popular tool especially used in the eco-design sector. Its methodology has been standardized by ISO 14 040 following four successive and iterative steps: goals and scope definition, life cycle inventory (LCI), impact assessment, and interpretation of results [16]. Thereafter, IAPs have been performed to estimate the environmental impacts directly generated by WA production.

2.3.2. Life Cycle Inventory Analysis and Modeling Assumptions

Figure 1 and Table 2 show respectively the main processes and flows related to the production of 1 ton of WA in BPP. They were inventoried based on Ecoinvent v.3.3. (2016)

database, Zurich, Switzerland [36] and several assumptions reflecting the current average situation in Brittany, especially regarding the three main aspects of the wood energy sector: wood fuel, energy production, and wood ash.

**Table 2.** Inventory of the main inputs and outputs related to the production of 1 ton of wet wood ash produced in a BPP.

| Flow/Process | Amount | Unit | Ecoinvent v.3.3 (2016) |
|---|---|---|---|
| **Inputs** | | | |
| Wet wood chips | 76.5 | t | – |
| Manual tree felling, forest * | 127.1 | h | Power sawing, with catalytic converter {RER} \| processing \| Alloc Rec, U |
| Wood chipping * | 13.3 | h | Wood chipping, chipper, mobile, diesel, at forest road {RER} \| wood chipping, mobile chipper, at forest road \| Alloc Rec, U |
| Dried wood chips transport (to BPP) | 76.5 × 50 = 3825 | tkm | Transport, freight, lorry 7.5–16 metric ton, EURO5 {RER} \| market for \| Alloc Rec, U for 50 km |
| Water | 250 | kg | Tap water {RER} \| market group for \| Alloc Rec, U |
| Electricity | 3301 | kWh | Electricity, medium voltage {FR} \| market for \| Alloc Rec, U |
| **Outputs** | | | |
| Heat * | 254.7 | MWh | Heat, district or industrial, other than natural gas {RER} \| heat production, hardwood chips from forest, at furnace 5000kW \| Alloc Rec, U |
| Wet wood ash | 1 | t | - |

* Process adapted to the study context in the Ecoinvent database.

The present study considers a combustion process unfolding in a BPP with an installed capacity higher than 1 MW, using grate furnace technology and an average efficiency estimated at 90% [37]. Note that co-generation plants were not considered since 95% of the BPPs in Brittany are strictly dedicated to heating. Forest wood chips, as the first resource used in the studied region, were considered as fuel with 1% total ash and 30% moisture content. The wood chips supply chain was described by CNPF (National Center of Forest Property) illustrating a common French scenario for forest cleaning operations [38]. Technical data related to the different processes involved were recovered from [39]. At present, trees are manually cut down with a chainsaw with a productivity of 0.94 ton·h$^{-1}$ and a fuel consumption about 1.42 kg·ton$^{-1}$. Then, wood is chipped onsite with a chipping facility (9 ton·h$^{-1}$), consuming 0.31 kg of fuel per ton of chipped wood. Finally, the wood chips are transported to the drying/storage platform, where a final mass loss of 36% after drying is assumed [38].

A low heating value of 3.4 kWh per kg of wood fuel was assumed with an average process efficiency estimated at 90% [40]. Data related to emissions and electricity consumption of the biomass boiler were recovered from the Ecoinvent database. In this regard, a consumption of 0.0036 kWh of electricity is estimated per MJ of heat produced, considering the French electrical mix as for all other electricity consuming processes (reference year 2012 in the Ecoinvent v3.3 database). At the end of the combustion process, a wet collection of WA is modeled, as it is the main method of WA collection for BPPs with an installed capacity higher than 1 MW, with up to 25% moisture content in WA [4]. In consequence, the actual amount of dried WA contained in 1 ton of wet WA produced is about 750 kg; hence, 250 kg of water is used to moisturize it. In order to determine the main inputs and outputs of the combustion process (wet/dried WA, water, wet/dried wood chips, electricity, heat, etc.), calculations were realized in the reverse order of the process, from ash production to wood chip production. Thus, 76.5 tons of dried wood chips are necessary to produce 1 ton of wet WA.

Regarding transportation of wood chips, road transport was assumed to be performed by 7.5–16 tons lorries complying with EURO5 standards [41]. The supply transport includes two specific components: transport from logging sites to drying/storage platforms, and final transport to the BPP. Due to lack of data on transport from logging sites, only the final transport of dried wood chips was taken into account. Smaller BPPs stock up within a radius smaller than 50 km. Nonetheless, larger BPPs often have a multitude of suppliers for an average transport distance within a radius of less than 100 km [42]. Given the size of the considered BPP (more than 1 MW), an inter-regional supply of 50 km is considered [43]. Note that infrastructure construction, combustion of additional energy resources (e.g., natural gas), and maintenance of boilers were excluded from the modeling.

### 2.3.3. Life Cycle Impact Assessment

Data were recovered from the Ecoinvent v.3. database [36], for impact assessment of the different processes considered. In case of WA production, air quality, energy, and natural resources consumption are crucial issues. To address these challenges, 11 environmental indicators were calculated with SimaPro 9.0.0.49 software, Amersfoort, The Netherlands [44], using the two internationally accepted methods CML-IA Baseline v.3.04/EU25 (2013) [45]: abiotic depletion, depletion of abiotic resources (fossil fuel), climate change (GWP100a), ozone layer depletion, photochemical oxidation, human toxicity, freshwater and marine aquatic ecotoxicity, terrestrial ecotoxicity, acidification and eutrophication. Only midpoint indicators focusing on single environmental impacts were considered in this study as a result of the higher level of uncertainty of endpoint models compared to midpoint models [46]. In fact, it was proved that the endpoint approach should ultimately be used as a supplement [47].

### 2.3.4. Impact Allocation Procedures

Commonly, only one type of IAP is used in studies. However, IAP have a large influence on a system's environmental burden, especially regarding co-product allocation [48,49]. Consequently, this study explored different allocation methods to assess the environmental burden of WA from BPP combustion. These allocation methods are based on a general concept as follows (Equation (3)) [18].

$$\vec{F}_{\text{product/co-product}} = C \times \vec{F}_{\text{global process}} \tag{3}$$

With:

- $\vec{F}_{\text{product/coproduct}}$ and $\vec{F}_{\text{global process}}$ referring, respectively, to the flow inventories related to the environmental burden of product or co-product(s) and those related to the global process
- C referring to the allocation coefficient that varies according to the allocation method considered (Equations (4a)–(4c))

Mass allocation coefficient:

$$C_{\text{m}} = \frac{M_{cp}}{M_p + M_{cp}} \tag{4a}$$

Energy allocation coefficient:

$$C_{\text{en}} = \frac{E_{cp}}{E_p + E_{cp}} \tag{4b}$$

Economic allocation coefficient:

$$C_{\text{ec}} = \frac{€_{cp} \cdot A_{cp}}{€_p \cdot A_p + €_{cp} \cdot A_{cp}} \tag{4c}$$

With:

- $M_p$ and $M_{cp}$, respectively, referring to the mass of product (energy in this case) and the sum of co-product masses produced during the process (only WA in this case)
- $E_p$ and $E_{cp}$, respectively, referring to the energy quantity contained in the product and co-product(s)
- €$_p$ and €$_{cp}$, respectively, referring the sum of the price per unit of product and co-product(s)
- $A_p$ and $A_{cp}$, respectively, referring to the amount of product and co-product. In our case, $A_p$ is the energy quantity produced by the BPP ($E_p$) and $A_{cp}$ is the mass of ash produced by the BPP ($M_{cp}$), for the same quantity of heat generated ($E_p$).

Table 3 inventories all the values used to calculate the mass, energy and economic allocation coefficients. Regarding this case study, the main product is heat and the only co-product considered is WA with no distinction by type of WA (fly and bottom ash). The different allocation procedures were carried out considering the modeling assumptions previously described for the combustion of 1 ton of dried wood chips (wood chips with 1% ash and 30% moisture content, 3.74 kWh·kg$^{-1}$ of wood chips, etc.). Note that the heat is assumed to not have a mass just as WA has no energy content. In case of the economic allocation, different prices were considered for the heat assumed to be produced in urban heating networks as well as for WA, based on its management from agricultural spreading to landfilling. Indeed, for a BPP manager, the cheapest option to get rid of the ash is to spread it in fields, but he still must account for the storage, manipulation, and the fuel for the tractors [50]. The lowest estimate cost for this operation would be around 60 €·ton$^{-1}$ of ash. If he wants to dispose ash into landfill, as it can be considered as dangerous matter, the cost of the operation can go up to 400 €·ton$^{-1}$ [51]. There is as yet no benefit for the BPP manager to dispose of the ash, only costs.

**Table 3.** Values used to calculate the mass, energy, and economic allocation coefficients for 1 ton of WA (p refers to the main product of the BPP—the energy and cp refers to the co-products of BPP—the WA).

|  | Unit | Value | Reference |
|---|---|---|---|
| $M_p$ | ton | 0 | - |
| $M_{cp}/A_{cp}$ | ton | 1 | From calculations based on assumptions used in Section 2.2 |
| $E_p/A_p$ | MWh | 254.7 | |
| $E_{cp}$ | MWh | 0 | - |
| €$_p$ | €·MWh$^{-1}$ | 51 to 96 | [50] |
| €$_{cp}$ | €·ton$^{-1}$ | −30 to −400 | [51] |

The allocation coefficient can be calculated with the flow inventories considering only raw materials flows upstream from the global process. These flow inventories can be converted into downstream environmental impacts to assess the environmental burden of the product or co-product, as expressed by Equation (5).

$$\vec{I}_{\text{product/co-product}} = C \times \vec{I}_{\text{global process}} \tag{5}$$

With:

- $\vec{I}_{\text{product/co-product}}$ and $\vec{I}_{\text{global process}}$ referring, respectively, to the environmental impacts of the product or co-product and those related to the global process.

## 3. Results and Discussion

### 3.1. Estimation and Location of Wood Ash Deposits

Estimations of Brittany's total WA tonnages produced per year are given in Figure 2. In this region, BPPs produced between 2800 to 8900 tons of WA in the reference year 2017 (a) depending on the hypothesis considered. In the event that SCRCAE targets for the 2050 (b) horizon are met [21], tonnages could potentially increase by more than 75% with 5000 to 15,700 tons of ash generated. Note that these results reflect optimum tonnages of WA produced in the studied area and can, therefore, be subject to uncertainties. In other words, they can be lower in reality due to several factors influencing WA production (weather, variability of wood fuel, resource availability, evolution of BPP projects, etc.). As regards the assumptions made regarding the type of boiler and the related ash production by type, fly ash tonnages are unsurprisingly estimated lower than bottom ash ones in both regions. Conclusions can be different with alternative hypotheses; however, they can provide an order of magnitude of the ash deposit in Brittany nowadays and in the future. Moreover, the inherent features of ash differ according to the type considered and so can be an advantage or obstacle to their use into a specific industrial sector.

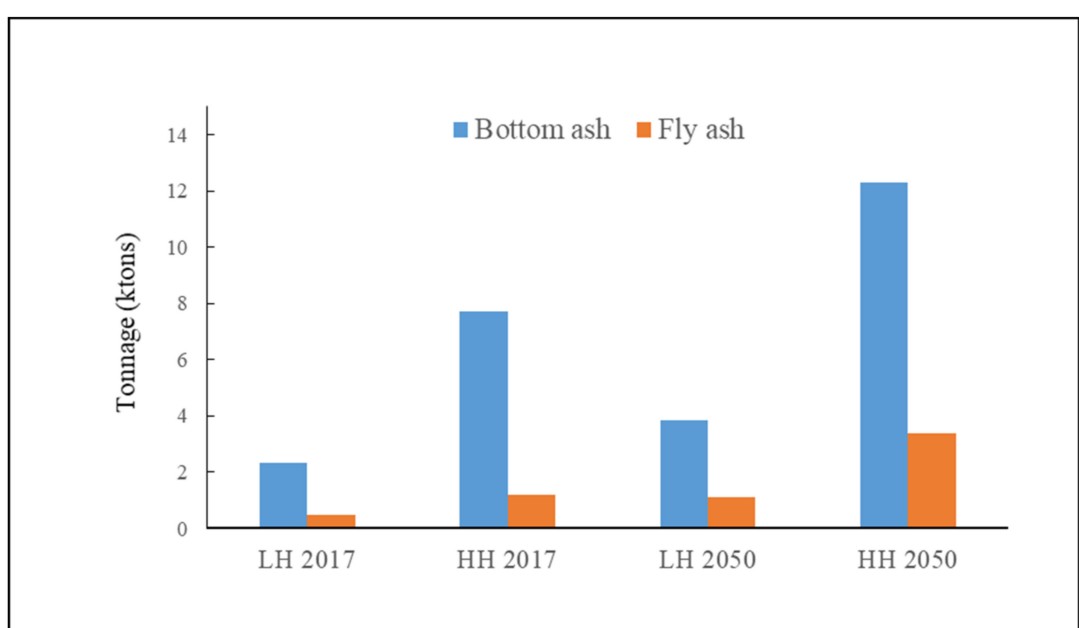

**Figure 2.** Global wood ash deposits in Brittany for 2017 reference year and projections for 2050. Note that LH refers to "Low Hypothesis" and HH to "High Hypothesis" from Table 1.

### 3.2. Life Cycle Assessment Results and Discussion

Table 4 and Figure 3 present the results of the impact assessment and contribution analysis or production of 1 ton of wood ash (WA) in a BPP for the reference year of 2017.

#### 3.2.1. Environmental Performance of Biomass Power Plants

As shown in Figure 3, to produce 1 ton of WA and 254. MWh of heat, three main hotspots have been identified i.e., wood chips combustion, wood chips production, and transport of the supplies. The wood chips production process makes the largest contribution of the environmental impact categories (8 over 11 categories): terrestrial ecotoxicity (64%), abiotic depletion (fossil fuel) (57%), global warming (52%), photochemical oxidation (52%), ozone layer depletion (49%), fresh water ecotoxicity (45%), abiotic depletion (36%), and marine aquatic ecotoxicity (35%). This is mainly due the chippers that consume large amounts of gasoline, as the tree growth was not taken into account as biogenic carbon. This is discussed further in this work. The wood combustion process is the second major contributor in most of the different categories of impacts, as ozone layer depletion (17%),

fresh water and marine aquatic toxicity (19 and 21% respectively), and photochemical oxidation (32%); and it also the main contributor when looking at eutrophication (75%) and acidification (74%), as well as human toxicity (66%). Energy production processes (energy conversion to produce heat and electricity) and the use of multiple facilities (boiler, ash collecting facilities) are the major contributors.

**Table 4.** Environmental impacts of wood ash based on the mass allocation method.

| Calculation Methods | Impact Categories | Units | Impacts of the Whole BPP, Producing 1 ton of WA and Energy | Impacts of 1 ton of Wet WA with the Mass Allocation Method (Cm = 1.3%) |
|---|---|---|---|---|
| CML-IA Baseline v.3.04 (2013) | Abiotic depletion | kg $Sb_{eq}$ | $1.10 \times 10^{-2}$ | $1.43 \times 10^{-4}$ |
| | Abiotic depletion (fossil fuel) | MJ | $9.26 \times 10^{4}$ | $1.20 \times 10^{-3}$ |
| | Global warming (GWP 100a) | kg $CO_{2eq}$ | $6.87 \times 10^{3}$ | 89.3 |
| | Ozone layer depletion | kg $CFC-11_{eq}$ | $1.31 \times 10^{-3}$ | $1.71 \times 10^{-5}$ |
| | Human toxicity | kg $1,4-DB_{eq}$ | $4.22 \times 10^{3}$ | 54.8 |
| | Freshwater aquatic ecotoxicity | kg $1,4-DB_{eq}$ | $8.71 \times 10^{2}$ | 11.3 |
| | Marine aquatic toxicity | kg $1,4-DB_{eq}$ | $1.71 \times 10^{6}$ | $2.22 \times 10^{4}$ |
| | Terrestrial ecotoxicity | kg $1,4-DB_{eq}$ | 92.1 | 1.20 |
| | Photochemical oxidation | kg $C_2H_{4eq}$ | 18.3 | $2.37 \times 10^{-1}$ |
| | Acidification | kg $SO_{2eq}$ | 92.5 | 1.20 |
| | Eutrophication | kg $PO_{4eq}$ | 24.5 | $3.18 \times 10^{-1}$ |

**Eq**: equivalent; **BPP**: Biomass Power Plant; **WA**: wood ash.

The impact of transport from drying/storage platforms to the BPP ranges from 1% to 33%. Considering a local supply (50 km), transport contributes to less than 5% of many impact categories (human toxicity, photochemical oxidation, acidification, eutrophication, and terrestrial ecotoxicity). Its contribution is, however, higher for global warming (14%), ozone layer depletion (13%), freshwater and marine ecotoxicity (11 and 17%), abiotic depletion (33%), and abiotic depletion (fossil fuel) (16%) due to the production of the vehicle fuel. The impact of water consumption was negligible in all impact categories.

The biggest challenges for the development of the biomass energy industry may be global warming and energy resource consumption [37,52]. In this regard, here we focus on two impact categories: global warming potential and energy resource depletion (abiotic depletion, especially fossil fuel). Regarding the global warming impact, the results are similar to the literature data for BPPs using a local wood supply chain to produce heat. Considering all stages of the forest-wood supply chain and the energy conversion phase, this study estimates that 8.85 t $CO_{2eq}$ are released per ton of WA generated which equals to 38.6 g $CO_{2eq}$/kWh produced, whereas 14–41 kg $CO_{2eq}$/kWh are reported in other French studies [39,53,54]. Several factors can explain the difference in results. First, it can be noted that the contribution of wood combustion processes to global warming (12%) is lower than expected, which is partially due to the mandatory use of dust collectors and electrostatic precipitators in large BPPs ($\geq$1MW); this results in the reduction of greenhouse gas emissions playing a key role in global warming. Consequently, the contribution of other impacting processes (wood chip production and transport) becomes higher. Moreover, the fact that transport of chipper and wood chips to/from the logging site have not been considered may also have a non-negligible repercussion on results [55]. Note that $CO_2$ emissions from biogenic carbon have been neglected in this study; thus, the overall global warming impact may be reduced. In fact, if the biogenic carbon were considered, the BPP impacts might increase, becoming equivalent to, or even higher than, the impacts of a heating plant using natural gas [56].

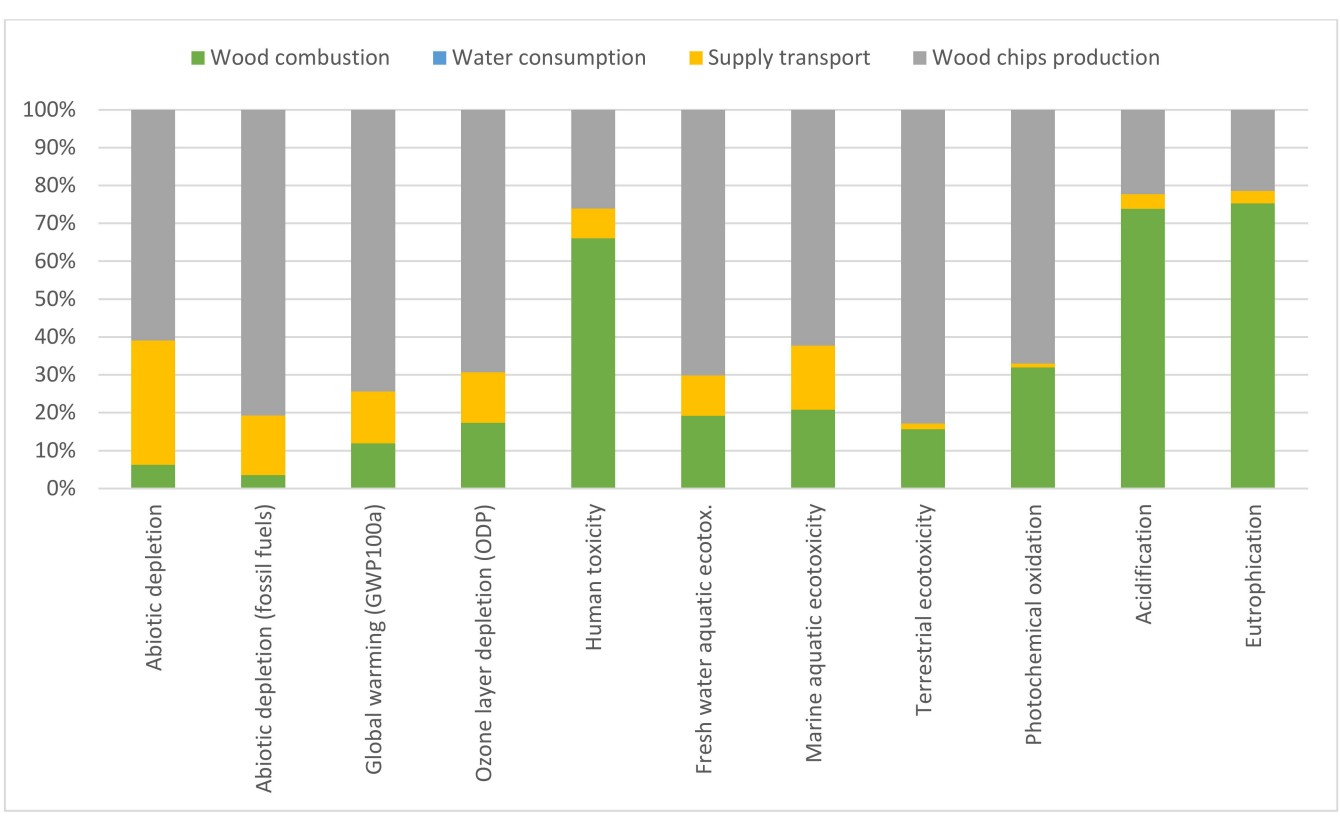

**Figure 3.** Impacts of 1 ton of wood ash from the combustion of wood chips in a 5 MW BPP located in Brittany, using CML Baseline.

As some impacts can be generated by the wood chip production, a short sensitivity analysis has been carried out on the type of wood used to make the wood chips. The presented results consider only the operation to transform the wood biomass within chips. The Ecoinvent process of wood chips was thus used to compare the results. When using the oak wood chips (wood chips, wet, measured as dry mass (DE) hardwood forestry, oak, sustainable forest management Alloc Rec, U), the results would decrease by 6% to 55% on the different impact categories, and especially by 38% in the global warming potential. Indeed, the prime assumption made was to exclude the biogenic carbon as the most frequent assumption in LCA. This carbon ($CO_2$) is captured and stored during the biomass growth, and it is assumed than all of it is released during the end-of-life of the biomass (the burning emissions in our case). Therefore, the total of input/output of biogenic $CO_2$ can be negligible over the whole biomass lifetime. On top of this analysis, the CML IA baseline method used to calculate the emissions does not have any characterization factor to consider properly the biogenic carbon.

Using wood chips made from oak, other tree species were then carried out as wood chips (beech, pine and spruce), as well as bark chips (hard and soft wood). The results demonstrated that wood chips made from other tree species would increase the global LCA results from 0.1–4.1% (beech) to 2.0–14.9% (spruce) whereas bark chips trend to decrease the previous results by −3.2% to −16.2% (soft and hard wood). This highlights the importance of describing the biomass type used in BPPs as it can highly modify the results.

The size of a BPP, reflected by its installed power capacity (<1 MW or >1 MW), has a strong influence on impacts generated by transport of the supplies. As reported by Paletto et al., (2019), in case of larger BPPs the supply distance tends to be higher, compared to smaller BPPs, for which foreign suppliers can be used. A longer distance of transport means a higher consumption of fuel, leading to a non-negligible increase of the overall impact [53]. In the case of wood chips coming from more than 250 km further away (rather than 50 km as used in the previous calculations), overall impact values can

increase from 5% to 95%. Global warming (+61%), ozone layer depletion (+95%), freshwater ecotoxicity (+76%), and non-renewable resource depletion (+72%) are unsurprisingly the most impacted environmental indicators. In this regard, supply transport remains a key parameter to consider in the BPP project design phase. These conclusions are in agreement with multiple authors who pointed the importance of short wood supply chains to reduce the environmental but also economic impacts of the forestry industry [57–59].

### 3.2.2. Environmental Burden of Wood Ash

Due to their very different natures (heat and ashes) the energy ($C_{en}$) and mass allocation ($C_m$) methods are not directly suitable for our study, as ash has no energy content and heat, no weight. However, the energy and the mass are crucial to determine the economic allocation ($C_{ec}$), based on recent values.

For the mass allocation, as we know the amount of wood burned (119.5 tons of wet wood chips) to produce 1 ton of WA, we could estimate that the burned part of the biomass was transformed into heat (main product) and other gaseous emissions; the remaining part of the wood is composed of the ashes. Using this data, we could indirectly estimate the $C_m$ as 1.3% with this method (1t of WA is generated from an input of 76.5t of wood). Using data from Table 1, for an average over Brittany region considering the different biomass mixes and their own ash content, and the fact that the BPP process requires 25% of moisture to collect them, the mass allocation ($C_m$) would vary between 1.4% (LH) to 2.4% (HH) for the whole region, not for specific BPPs. Concerning the different sources of biomass, the individual ratios could vary between 0.6% (LH for sawmill waste and for wood waste) to 2.5% (HH for forest wood chips and for wood waste). These values are in the same order of magnitude and, therefore, the Cm calculated for the described BPP will be applied.

Regarding the economic allocation, the potential value of WA on the market can differ from that used in the study which are based on its recycling (spreading) and elimination (landfilling). It may result in a possible variation of the allocation coefficient that can decrease in the case of lower prices and inversely can increase in the case of higher costs. All hypotheses were tested over this economic allocation and the allocation factor ($C_{ec}$) varied between −0.1% and −3.2% along with the highest/lowest price and cost estimates. Indeed, as the BPP manager must pay to get rid of the WA, these factors are mathematically negative. Therefore, it was decided not to consider them for estimations of the environmental burden of WA as it will artificially increase the environmental load of the BPP for an allocation which has no reality. Therefore, considering an economic allocation, the WA cannot be considered as a co-product but as waste from the heat production process.

Few studies deal with co-product allocation in the biomass energy production industry. Chen et al. (2010) have assessed the environmental burden of fly ash as co-products of coal combustion plants and granulated blast furnace slag [18]. Boschiero et al. (2015) is one of the rare studies that analyze the influence of allocation procedures for biomass co-products (woodchips from apple orchards) on the environmental performance of bioenergy production. They have shown that benefits or disadvantages related to the co-product depended on the type of allocation considered. Thus, considering an economic allocation for the wood chips from apple orchards guarantees a reduction of overall impacts with a significant reduction of greenhouse gases emissions (up to 97%) and primary energy demand (up to 97%) [43].

Most of the studies tend to avoid allocation due to uncertainties related to the different methodologies proposed. Others seem to focus on analyzing impact allocation procedures, mainly dealing with other biofuel production such as ethanol or methanol or other types of by-products [59–62].

From the different allocation methods presented in this study, only the mass methods could be used. It is estimated that up to 1.3% of total impacts generated by the combustion process in BPPs are allocated to WA (Table 4), and this ratio can be considered between 1.1% and 1.9% when considering all BPPs within the region of Brittany. Therefore, it can be concluded that WA cannot be considered as a co-product of heat generation from a

BPP, but as a waste product of this industrial process. No environmental burden should be attributed to the WA, especially the fly ash part of the WA (only 10% of the WA in gate-furnace BPP). To mitigate the hazard when disposed in landfill, fly ash can easily be incorporated within concrete formulations. Thanks to its chemico-physical properties, WA can be used as an alternative to cement within concrete formulation and thus reduce the environmental impacts of concrete production [63]. It would also be pertinent to locate these fly ash deposits in the vicinity of local concrete production factories. This would optimize transportation of the WA to concrete plants, and thus minimize the impacts of the alternative concrete mix [64].

Estimating the impacts of multifunctional processes (with more than one product per process) is a burning issue, especially to assess the potential for gains in the circular economy through valorizing waste streams. Other methodologies can be used such as the Consequential LCA using substitution, as well as the recent version of Ecoinvent database (consequential or allocation at point of substitution) that allows broadening of system boundaries to avoid IAP [65,66]. However, the purpose of this study was to estimate the amounts of WA available in a specific region and decide if WA can be considered as a co-product or as a waste product of energy production. Besides, consequential LCA (CLCA) requires a complete picture of the valorization scenarios (i.e., precise percentage of cement substitution by WA) and it would provide results corresponding to the specific set of parameters used in this CLCA (and its sensitivity analysis), whereas the purpose of the current work was to assess whether WA is a co-product of heat generation or its waste, using a screening ALCA, to be used within different valorization development work. The latter proposition (waste) was retained from the calculations and for further work using WA as a resource. When a precise scenario using WA as a resource would be set, it will be interesting to carry out an CLCA taking into consideration the "cradle to grave" boundaries, and/or comparing the different options for WA treatments, as developed by other studies [19,20].

## 4. Conclusions

The use of biomass to generate energy has grown significantly over the past decades and current projections are optimistic about a further development of the domain [67]. However, even if this industry provides an eco-friendly and sustainable energy source, it also produces a growing amount of waste, wood ash, that can be categorized depending on the technology used for heat production and on its chemico-physical properties. This waste usually goes to landfill (some ash is used as forest fertilizer). A growing amount of ash will be generated in the coming decades, even if there are negative consequences for human health, leading to an issue of handling the ash. It thus becomes interesting to find alternative uses for the ash, so it can become a resource for alternative industries and no longer just a waste product that needs to be buried. This raises the issue of the allocation of environmental impact between the heat produced in the BPP and the co-product, namely the WA. Quantitative estimates relating to environmental burden are crucial to provide a basis for further work, which might consider the use of WA as a resource in another industrial process.

The present study describes an approach to assessing resource implications of production of wood ash within a particular French region: Brittany. Cooperation between local wood energy experts (AILE and Fibois experts) in the studied area and academia (environmental experts) helped to strengthen the methodology implemented, by using verified data and standardized methods (Life Cycle Assessment, Impact Allocation Procedure). Moreover, the analysis of wood ash deposits permitted an increase in the stakeholders' knowledge, which can be shared not only with the technical public but also the non-technical public. The defined method can be replicated for other resources in a different geographical context, considering the specific goals and requirements. Nonetheless, this study is an isolated example of the use of the method in the literature. More studies are necessary to better understand its advantages but also limitations. One of the limitations

identified in the current method is the fact that assessing resources systematically requires a huge amount of good quality data that has to be verified by experts. In this way, technical and environmental competences can be coupled to further develop the sustainability of the industrial sector.

Awareness of the importance of estimating the quantities and the environmental burdens of wood ash, especially fly ash, is crucial as it can be used by industry as an alternative ingredient alongside cement. As presented in the LCA results, the type of wood used for the biomass and the transportation of this biomass to the BPP have crucial consequences for the BPP environmental assessment. The type of wood (form and origin) determines the composition of the ash generated from combustion and thus its potential usage within another industry. It has also been demonstrated through this study that the allocation of wood ash compared to energy production is so low that it can be negligible, and even more so when focusing on the fly ash part of wood ash production (10%). Therefore, wood ash cannot be considered as a co-product of heat generation, but as a waste product derived from this process. If opportunities of ash valorization appear and a commercial value is attributed to it, the status of ash as waste should be re-assessed through a similar study. However, due to market price fluctuations, the results may vary according to supply and demand, which is a great limitation for this type of evaluation. No environmental impact can be attributed to ash within the BPP boundaries. If used in the building industry, its impact would start to be assessed from the collection of the ash and its transportation. If local geographical scale of operations is ensured through bringing this ash to a local concrete factory, this should permit reduction of the environmental impacts of concrete, as Portland cement production is responsible for nearly 7% of global GHG emissions. Thus, location of wood ash deposits should be chosen to optimize the introduction of wood ash and fly ash within other industries, such as concrete production.

**Author Contributions:** K.D. and M.M.-C. made substantial contributions to the design of the work, performed the analysis and wrote the manuscript. V.A. and T.H. participated as specialists in material flow and waste management and regulation, respectively. H.D. supervised the study, and participated in the editing of the manuscript. All authors have read and agreed to the published version of the manuscript.

**Funding:** This research was funded by ADEME, France (Environmental French Agency) which financed the BIMGC (Valorisation des cendres de chaudière biomasse dans l'élaboration de matériaux composites pour le genie civil, Valorization of wood biomass ash in the development of composite materials for civil engineering) project (2018–2021).

**Institutional Review Board Statement:** Not applicable.

**Informed Consent Statement:** Not applicable.

**Data Availability Statement:** Not applicable.

**Acknowledgments:** The authors thank Marie-Anne Hairan (UniLasalle-EME) for her helpful comments for improving the English.

**Conflicts of Interest:** The authors declare that they have no known competing financial interest or personal relationships that could appear to influence the work reported in this article.

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
