# Peer review of "Quantification and Environmental Assessment of Wood Ash from Biomass Power Plants: Case Study of Brittany Region in France"

_sustainability, doi:10.3390/su14010099_

Round 1

Reviewer 1 Report

The well-structured article focuses on an interesting and relevant topic: wood ash quantification and application of LCA to  assess its environmental impact. However, the study is quite limited and flawed in its methodology, especially concerning the allocation procedures which has a major influence on the outcme. Hence, I advise to reject the manuscript for now. There is some potential but a very thorough reconsideration is needed.

Major comments:

  • In general, distinguish clearly between impact of “waste” and impact of “waste handling/treatment”. This should be consistent and clear throughout the manuscript. For example in the abstract is written: “ IAP analysis pointed out that the environmental burden of WA from combustion process in BPP is low”. Is this then for the waste itself or the treatment? This is not clear.
  • While the article focuses on a multifunctionality and allocation procedures, which are both relevant for waste handling, the actual methodological quality and consideration of literature are poor. The whole aspect of attributional versus consequential LCA in the consideration of methodological choices is overlooked (Ekvall and Weidema, 2004; Schaubroeck et al., 2021a, 2021b, Schrijvers et al., 2016a, 2016b). The choice of a procedure will depend on the goal of the study. Nowhere has been specified what the actual goal is, which is though the first goal according to ISO 14040-14044. (I know that this is a challenging topic, but if the authors focus on it). If you want to keep the results at is, it is better to consider your work as an Attributional LCA, since you only apply allocation/partitioning.
  • You mention: “However, due to recurrent difficulties to expand the system in case of by-product, other types of allocation based on physical and economic properties of the by-product are mostly used today [29].” This is highly questionable. There is now a complete version of the ecoinvent database that is consequential and many consequential LCA studies have been conducted using substitution. Why have you not considered this alternative?
  • Related with the multifunctionality issue, the whole dilemma of whether wood ash is a product or waste flow is overlooked. From the methodological choices, I understand that wood ash is considered as a product. Yet, if it is considered a waste stream, than the multifunctionality issue should be solved differently. See literature on this matter (Schaubroeck et al., 2021a). Moreover, the fact that it has a negative cost (as shown in Table) means that it may be rather considered a waste flow. This relates with the complete lack of specification of the goal of the study related with this matter.
  • It has not been specified which exact version of ecoinvent is considered. This is very crucial as it should best align with the methodological choices for the foreground system, most notably the allocation procedures on which your study focuses.
  • In table 4 mass, energy and economic are mentioned in the caption, but only energy and mass are shown?! The energy content is also 0 so what is the point?!
  • Eventually only two allocation coefficients are considered (mass and energy) from which the energy is 0. This scope is too limited whereas the focus is also on this topic.
  • In table 3, the mass value is 0, or is this for the heat? Please distinguish between heat and wood ash as a product
  • I have stopped listing further major comments as there are already too many crucial ones that should be first handled.

Minor comments

You wrote: “ Boulday and Marcovecchio (2016)”. Please adapt citation style to that of the journal

You mention: “Biomass is renewable and abundant in the nature.”. This is a bit a too simplistic and general statement. “abundant” is relative and so is “renewability”. Please leave out or describe better.

There are many language issues. Here, I list a few. Please have the English language checked.

  • You wrote: “ spreading to as nutrient” check language of sentence.
  • You wrote: “Because of its physical structure of fine partic-ulate matter”. Should be “the physical structure, but sentence should also be checked.

You mention: “By limiting some of the environmental impacts of wood energy production, it is favorable to shifting to renewable energy and can also contribute to returning biological resources to the Biosphere [9].” This is quite a bold statement and you only cite a report of Ellen MacArthur foundation? Please consider detailed studies on this matter (Gibon et al., 2017b, 2017a; Luderer et al., 2019) and reconsider if necessary.

Ekvall, T., Weidema, B.P., 2004. System boundaries and input data in consequential life cycle inventory analysis. Int. J. Life Cycle Assess. 9, 161–171. https://doi.org/10.1007/BF02994190

Gibon, T., Arvesen, A., Hertwich, E.G., 2017a. Life cycle assessment demonstrates environmental co-benefits and trade-offs of low-carbon electricity supply options. Renew. Sustain. Energy Rev. 76, 1283–1290. https://doi.org/10.1016/j.rser.2017.03.078

Gibon, T., Hertwich, E.G., Arvesen, A., Singh, B., Verones, F., 2017b. Health benefits, ecological threats of low-carbon electricity. Environ. Res. Lett. 12, 034023. https://doi.org/10.1088/1748-9326/aa6047

Luderer, G., Pehl, M., Arvesen, A., Gibon, T., Bodirsky, B.L., de Boer, H.S., Fricko, O., Hejazi, M., Humpenöder, F., Iyer, G., Mima, S., Mouratiadou, I., Pietzcker, R.C., Popp, A., van den Berg, M., van Vuuren, D., Hertwich, E.G., 2019. Environmental co-benefits and adverse side-effects of alternative power sector decarbonization strategies. Nat. Commun. 10, 1–13. https://doi.org/10.1038/s41467-019-13067-8

Schaubroeck, T., Gibon, T., Igos, E., Benetto, E., 2021a. Sustainability assessment of circular economy over time: Modelling of finite and variable loops & impact distribution among related products. Resour. Conserv. Recycl. 168, 105319. https://doi.org/10.1016/j.resconrec.2020.105319

Schaubroeck, T., Schaubroeck, S., Heijungs, R., Zamagni, A., Brandão, M., Benetto, E., 2021b. Attributional & Consequential Life Cycle Assessment: Definitions, Conceptual Characteristics and Modelling Restrictions. Sustainability 13, 7386. https://doi.org/10.3390/su13137386

Schrijvers, D.L., Loubet, P., Sonnemann, G., 2016a. Critical review of guidelines against a systematic framework with regard to consistency on allocation procedures for recycling in LCA. Int. J. Life Cycle Assess. 1–15. https://doi.org/10.1007/s11367-016-1069-x

Schrijvers, D.L., Loubet, P., Sonnemann, G., 2016b. Developing a systematic framework for consistent allocation in LCA. Int. J. Life Cycle Assess. 1–18. https://doi.org/10.1007/s11367-016-1063-3

Author Response

Dear Editor,

Thank you for your mail and for the expert Reviewer ‘comments. Please find enclosed the revised manuscript. The points raised by the Referees are clarified below and the manuscript has been revised accordingly.

 Response to the Reviewer’s comments:

 Reviewer 1

 Major comments:

  • In general, distinguish clearly between impact of “waste” and impact of “waste handling/treatment”. This should be consistent and clear throughout the manuscript. For example in the abstract is written: “ IAP analysis pointed out that the environmental burden of WA from combustion process in BPP is low”. Is this then for the waste itself or the treatment? This is not clear.

Answer: This study is focusing only on the waste for the BPP, up to the door of the plant. The handling/treatment of them have been studied within another publication, with their transportation and the integration in another process. Some discussion has been added within the introduction to make it clearer and the abstract has been reviewed in that perspective.

  • While the article focuses on a multifunctionality and allocation procedures, which are both relevant for waste handling, the actual methodological quality and consideration of literature are poor. The whole aspect of attributional versus consequential LCA in the consideration of methodological choices is overlooked (Ekvall and Weidema, 2004; Schaubroeck et al., 2021a, 2021b, Schrijvers et al., 2016a, 2016b). The choice of a procedure will depend on the goal of the study. Nowhere has been specified what the actual goal is, which is though the first goal according to ISO 14040-14044. (I know that this is a challenging topic, but if the authors focus on it). If you want to keep the results at is, it is better to consider your work as an Attributional LCA, since you only apply allocation/partitioning.

Answer: The goal of the LCA has been added in the introduction and in the part 2.3.1 and indeed we are doing an attributional LCA. Indeed, as the WA are intended to be incorporated within concrete mix (following LCA work that will be published later), we followed the European standards for construction EN 15978 that only refers to Attributional LCA. (that have been precised in the section 2.3.1).

  • You mention: “However, due to recurrent difficulties to expand the system in case of by-product, other types of allocation based on physical and economic properties of the by-product are mostly used today [29].” This is highly questionable. There is now a complete version of the ecoinvent database that is consequential and many consequential LCA studies have been conducted using substitution. Why have you not considered this alternative?

Answer: You are completely right, this is obsolete (from 2013) and scientific work has improved it in the last decade. However, we did not want this paper to focus on one or a couple of substitution products, but to be more general for the use of wood ash (in the building sector but maybe also in other industrial sectors). Besides, not all wood ash can replace cement in the concrete mix, it would be highly dependent on the biomass type used to run the BPP. This sentence has been removed.

  • Related with the multifunctionality issue, the whole dilemma of whether wood ash is a product or waste flow is overlooked. From the methodological choices, I understand that wood ash is considered as a product. Yet, if it is considered a waste stream, than the multifunctionality issue should be solved differently. See literature on this matter (Schaubroeck et al., 2021a). Moreover, the fact that it has a negative cost (as shown in Table) means that it may be rather considered a waste flow. This relates with the complete lack of specification of the goal of the study related with this matter.

Answer: Indeed, when working on the economic allocation between heat and WA, it appears well that WA cannot be considered as a coproduct but remains as a waste. The goal of the study has been precised in the section 2.3.1 and conclusions were modified to make this statement clearer.

  • It has not been specified which exact version of ecoinvent is considered. This is very crucial as it should best align with the methodological choices for the foreground system, most notably the allocation procedures on which your study focuses.

Answer: The version of Ecoinvent was detailed in the section 2.3.2.

  • In table 4 mass, energy and economic are mentioned in the caption, but only energy and mass are shown?! The energy content is also 0 so what is the point?!

Answer: Sorry, the caption has not been updated indeed. This table has been completely corrected, only the results for mass allocation are available.

  • Eventually only two allocation coefficients are considered (mass and energy) from which the energy is 0. This scope is too limited whereas the focus is also on this topic.

Answer: You are right, the scope was too limited, it has been changed to a larger question: are wood ash a coproduct of heat production though BPP (with some environmental impacts to be attributed) or a waste of this energy sector (without environmental burden)?

  • In table 3, the mass value is 0, or is this for the heat? Please distinguish between heat and wood ash as a product

Answer: The caption has been improved since, with an explanation between the product (the heat) and the co-product (the wood ash).

  • I have stopped listing further major comments as there are already too many crucial ones that should be first handled.

Minor comments

You wrote: “ Boulday and Marcovecchio (2016)”. Please adapt citation style to that of the journal

Answer: Thank you for this comment, I made the correction in the revised manuscript.

You mention: “Biomass is renewable and abundant in the nature.”. This is a bit a too simplistic and general statement. “abundant” is relative and so is “renewability”. Please leave out or describe better.

Answer: I agree with this remark and I delete this sentence. The correction was made in the revised manuscript.

There are many language issues. Here, I list a few. Please have the English language checked.

  • You wrote: “ spreading to as nutrient” check language of sentence.
  • Answer: The correction was made.
  • You wrote: “Because of its physical structure of fine particulate matter”. Should be “the physical structure, but sentence should also be checked.

Answer: The correction was made.

You mention: “By limiting some of the environmental impacts of wood energy production, it is favorable to shifting to renewable energy and can also contribute to returning biological resources to the Biosphere [9].” This is quite a bold statement and you only cite a report of Ellen MacArthur foundation? Please consider detailed studies on this matter (Gibon et al., 2017b, 2017a; Luderer et al., 2019) and reconsider if necessary.

Reviewer 2 Report

Reviewer´s comments on manuscript sustainability 1471358

General

The presented study focuses on an interesting topic of environmental assessment of wood ash from biomass power plants with Brittany region serving as a case study. Manuscript is neatly written; its aims are well defined and well justified. Materials and Methods part is clear apart from the issue of future energetic mix which need additional explanation or justification. While well presented in general, some of the results seem unrealistic and deserve a double check with a subsequent change in study conclusions accordingly. Language needs improvement, especially Abstract and Introduction parts. In general, the manuscript is well suited for Sustainability journal and can make a fine contribution after a minor revision.

The authors can find detailed comments below:

Detailed assessment and queries

Language: needs substantial improvement both from grammar and style point of view (especially Abstract and Introduction). A revision by a native speaker is needed.

Introduction: Apart from language issues it is neatly and concisely written and manages to highlight the importance of the selected research topic sufficiently.

Materials and Methods

Part. 2.1: The sentence “Therefore, the biomass consumption raised up to 552 kton for a generation of 394 MWth.” Please be more specific if the biomass consumption relates to 2020 and if, reformulate the sentence.

Table 1 and elsewhere: Check for correct use of “.” as decimal separator instead of “,”.

Part 2.3.2 Please specify the year (years) to which the considered value of French energetic mix relates. If your calculations include projection to 2050 it would be feasible to discuss how the energy mix would look like at that time. In addition, please double check the value of 0.0036 kWh in relation to 1 MJ, as 0.0036 kWh represents 12.96 kJ only.

Table 2 and elsewhere: All abbreviations and symbols first appearing in a table or a figure should be explained in the caption.

Figure 2: Try to supply artwork with better resolution.

Results and Discussion

Part 3.2.1: Please double check the stated value of 4.84 kg CO2 equivalent per one ton of ash, there might be an error in your calculations. The relative difference between your results and that of other studies is very big and cannot be explained just by the fact you mention in the related discussion. Consider just the over 3000 km needed for dried chips transport stated in Table 2. Imagine an average diesel consumption of 15 l/100 km, corresponding roughly to 300-330 kg of diesel consumed and to around 850 kg of solely CO2 produced. With regard to this, check the obtained values of all other environmental indicators as well.

Conclusions are supported by the study results. However, it might be necessary to revise them if some of the results change during manuscript revision (see the comment related to part 3.2.1).

References: Please format the references according to MDPI requirements and supply DOIs where available.

Author Response

Dear Editor,

Thank you for your mail and for the expert Reviewer ‘comments. Please find enclosed the revised manuscript. The points raised by the Referees are clarified below and the manuscript has been revised accordingly.

 Response to the Reviewer’s comments:

 Reviewer 2

Comments and Suggestions for Authors

Reviewer´s comments on manuscript sustainability 1471358

General

The presented study focuses on an interesting topic of environmental assessment of wood ash from biomass power plants with Brittany region serving as a case study. Manuscript is neatly written; its aims are well defined and well justified. Materials and Methods part is clear apart from the issue of future energetic mix which need additional explanation or justification. While well presented in general, some of the results seem unrealistic and deserve a double check with a subsequent change in study conclusions accordingly. Language needs improvement, especially Abstract and Introduction parts. In general, the manuscript is well suited for Sustainability journal and can make a fine contribution after a minor revision.

The authors can find detailed comments below:

Detailed assessment and queries

Language: needs substantial improvement both from grammar and style point of view (especially Abstract and Introduction). A revision by a native speaker is needed.

Answer: The manuscript was revised by a native speaker.

Introduction: Apart from language issues it is neatly and concisely written and manages to highlight the importance of the selected research topic sufficiently.

Answer: Thank you for this remark, the abstract and introduction was corrected by a native speaker.

Materials and Methods

Part. 2.1: The sentence “Therefore, the biomass consumption raised up to 552 kton for a generation of 394 MWth.” Please be more specific if the biomass consumption relates to 2020 and if, reformulate the sentence.

Answer: The addition of the year was added in the revised manuscript.

Table 1 and elsewhere: Check for correct use of “.” as decimal separator instead of “,”.

Answer: The correction was made.

Part 2.3.2 Please specify the year (years) to which the considered value of French energetic mix relates. If your calculations include projection to 2050 it would be feasible to discuss how the energy mix would look like at that time. In addition, please double check the value of 0.0036 kWh in relation to 1 MJ, as 0.0036 kWh represents 12.96 kJ only.

Answer: There was a confusion in the text. It was assumed that 0.0036kWh of electricity is required to run the BPP and to produce 1 MJ of heat. This has been corrected in the text and the reference year (from the database) of 2012 has been added.

Table 2 and elsewhere: All abbreviations and symbols first appearing in a table or a figure should be explained in the caption.

Answer: Thank you for this observation, the correction was made in the revised manuscript.

Figure 2: Try to supply artwork with better resolution.

Answer: Thank you for this suggestion, we improve the quality of Figure 2.

Results and Discussion

Part 3.2.1: Please double check the stated value of 4.84 kg CO2 equivalent per one ton of ash, there might be an error in your calculations. The relative difference between your results and that of other studies is very big and cannot be explained just by the fact you mention in the related discussion. Consider just the over 3000 km needed for dried chips transport stated in Table 2. Imagine an average diesel consumption of 15 l/100 km, corresponding roughly to 300-330 kg of diesel consumed and to around 850 kg of solely CO2 produced. With regard to this, check the obtained values of all other environmental indicators as well.

Answer: Distances in LCA are expressed as tkm which is the multiplication of the weight transported (76.5 t of wet wood chips) by the distance (50 km). The aim of having energy from wood is to reduce the impacts on the environment, therefore a local production of wood was considered. It has been detailed within the table now.

But you were right, there was a mistake in our calculations. Comparing the results to literature helped to find the error and to correct it. ADEME study gives an average emission of 28 gCO2eq/kWh and we assumed that our BPP was producing 254.7 MWh to have 1 t of wood ash. In this new version of our calculations, we found a value of 29.9 gCO2/kWh which is fairly similar.

Conclusions are supported by the study results. However, it might be necessary to revise them if some of the results change during manuscript revision (see the comment related to part 3.2.1).

Answer: Indeed, the conclusions have been improved with the inclusion of more results and a global discussion around the limits of this study and perspectives that could be developed.

References: Please format the references according to MDPI requirements and supply DOIs where available.

Answer: The correction was made in the revised version of the manuscript.

Reviewer 3 Report

Dear authors, the paper is very interesting and I think that the combination of a potential waste with a renewable source is very promising. I see the structure of the paper very clear, with methodology described in a perfect way. I have no comments on this part.

  1. Please check all text, there are mistakes
  2. The structure of equations is not clear better 1,2,3 and not a,b,c
  3. abstract I think that there is low space for results. This part can be improved
  4. The novelty of the paper is not well identified than existing literature. Where is the element in which you have identified the need to write this research?
  5. WIthin biomass context, there are key-concepts that are recently published, https://doi.org/10.1038/s41598-020-80732-0, https://doi.org/10.3390/en14185661, https://doi.org/10.3390/su13137098
  6. Results, can be improved through a comparison with previous results in literature
  7. Results, must be proposed on a sensitivity analysis on critical variables
  8. What are the main limits of this work (please report all in the last section)

Author Response

Dear Editor,

Thank you for your mail and for the expert Reviewer ‘comments. Please find enclosed the revised manuscript. The points raised by the Referees are clarified below and the manuscript has been revised accordingly.

 Response to the Reviewer’s comments:

 Reviewer 3

Comments and Suggestions for Authors

Dear authors, the paper is very interesting and I think that the combination of a potential waste with a renewable source is very promising. I see the structure of the paper very clear, with methodology described in a perfect way. I have no comments on this part.

1. Please check all text, there are mistakes

Answer: Thank you for this remark and we checked the text and correct mistakes

2. The structure of equations is not clear better 1,2,3 and not a,b,c

Answer: The correction was made in the revised manuscript

3. abstract I think that there is low space for results. This part can be improved

Answer: Abstract has been greatly improved thanks to your remark.

4. The novelty of the paper is not well identified than existing literature. Where is the element in which you have identified the need to write this research?

Answer: When redefining the scope of the study, the novelty has been put forward as this work is tending to give a basis for futher study that would take into account wood ash as a resource.

5. WIthin biomass context, there are key-concepts that are recently published, https://doi.org/10.1038/s41598-020-80732-0, https://doi.org/10.3390/en14185661, https://doi.org/10.3390/su13137098

6. Results, can be improved through a comparison with previous results in literature

Answer: few similar results exist in literature and the main one (emission of CO2eq/kWh) has been compared to French references for BPP, and the main conclusions of the LCA were in accordance with similar studies.

As a novelty, it was not possible to find data about the impact allocation of wood ash in the literature, it has mainly been used as a waste.

7. Results, must be proposed on a sensitivity analysis on critical variables

Answer: a short sensitivity analysis has been carried on the main component of the LCA and discussion were added around the values found for the impact allocation procedure.

8. What are the main limits of this work (please report all in the last section)

Answer: Thank you for your remark. This has been modified and implemented within the last section of the paper.

Round 2

Reviewer 1 Report

The article has been improved in a certain direction; steps have been made forad. Still crucial issues remain. See major points below. Hence, a major revision still seems needed.

  • You mention in the reply that WA would be a waste, and not a product. Yet, of what are you then quantifying the LCA impact. Its treatment right? That is the functionality. If you apply cutoff, it means that the waste flow as such does not have an impact, only its treatment has. You mention in the abstract: “ The LCA conducted through this study gave an emission of 38.6 gCO2eq.kWh-1,”, but of what is that then the LCA? What is the Functional Unit? All the co-products? But you mentioned that you focus on WA and it is a waste? The core focus of the LCA still is off or it is not explained consistently (especially in the abstract)
  • You also still mention: “Due to their very different natures (heat and ashes) the energy (Cen) and mass alloca-tion (Cm) methods are not suitable for our study”. However, did you eventually not choose mass allocation when considering WA as a product.
  • What is still the point of considering economic and energy allocation, if you consider it as a waste. I would advise to better distinguish between (a) a scenario where it is considered a product and (b) one where it is considered a waste. If it is considered a product, in the first scenario, it is needed that you specify its functionality as a product or useage for other products. This should be done at least as plausible.
  • If indeed ALCA is apprehended then messages need to be aligned with it. If your ALCA study aims to address the consequences of decisions, ALCA can only be used as an approximation and it should be mentioned in the discussion that CLCA should be better apprehended. ALCA as such can still provide insights on the “responsibility” of goods.
  • On another matter, the aspects of biogenics and dynamics of carbon emissions on global warming calculations has only been minorly covered. Please discuss this more thoroughly at least.
  • Similar literature on the same topic is still not well considered, e.g. such as the work of Gaudreault et al. (2020) and Tosti et al. (Tosti et al., 2020). Please compare with these works and showcase the novelty compared to these.

Mind typos such as “global worming”

Gaudreault, C., Lama, I., Sain, D., 2020. Is the beneficial use of wood ash environmentally beneficial? A screening-level life cycle assessment and uncertainty analysis. J. Ind. Ecol. 24, 1300–1309. https://doi.org/https://doi.org/10.1111/jiec.13019

Tosti, L., van Zomeren, A., Pels, J.R., Damgaard, A., Comans, R.N.J., 2020. Life cycle assessment of the reuse of fly ash from biomass combustion as secondary cementitious material in cement products. J. Clean. Prod. 245, 118937. https://doi.org/10.1016/j.jclepro.2019.118937

Author Response

Comments and Suggestions for Authors

The article has been improved in a certain direction; steps have been made forad. Still crucial issues remain. See major points below. Hence, a major revision still seems needed.

  • You mention in the reply that WA would be a waste, and not a product. Yet, of what are you then quantifying the LCA impact. Its treatment right? That is the functionality. If you apply cutoff, it means that the waste flow as such does not have an impact, only its treatment has. You mention in the abstract: “ The LCA conducted through this study gave an emission of 38.6 gCO2eq.kWh-1,”, but of what is that then the LCA? What is the Functional Unit? All the co-products? But you mentioned that you focus on WA and it is a waste? The core focus of the LCA still is off or it is not explained consistently (especially in the abstract)

Answer: the aim of this study was to assess whether the WA could be considered as a coproduct of heat generation or as a waste of it. Therefore, the LCA was carried out only “from craddle to gate” only, so the waste treatment has been excluded. It will be carried out within a following work. This has been corrected within the text and the abstract, and the functional unit has been added.

  • You also still mention: “Due to their very different natures (heat and ashes) the energy (Cen) and mass alloca-tion (Cm) methods are not suitable for our study”. However, did you eventually not choose mass allocation when considering WA as a product.

Answer: Indeed it is not directly suitable, as the energy does not have any weight. However, it can be used indirectly with a ratio of WA produced/weight of wood needed to produce this amount of WA. We added a short comment on this aspect in the paper.

  • What is still the point of considering economic and energy allocation, if you consider it as a waste. I would advise to better distinguish between (a) a scenario where it is considered a product and (b) one where it is considered a waste. If it is considered a product, in the first scenario, it is needed that you specify its functionality as a product or useage for other products. This should be done at least as plausible.

Answer: the point was to demonstrate that if economic or mass allocation are applied (with a WA considered as a co-product of heat with a function as being used in concrete mix), the environmental burden of WA is negligible, especially for fly ash (10% of WA). Therefore, if considered as a coproduct or as a waste, no impact can be attributed towards the production of fly ash, and so few for all wood ash. That thus represent a non-negligible interest to find further purposes for WA, than landfill.

  • If indeed ALCA is apprehended then messages need to be aligned with it. If your ALCA study aims to address the consequences of decisions, ALCA can only be used as an approximation and it should be mentioned in the discussion that CLCA should be better apprehended. ALCA as such can still provide insights on the “responsibility” of goods.

Answer: The ALCA presented aims at providing a reference for the production of WA, and this one could be then used to assess the efficiency of different ways of valorisation of them (not only within a concrete mix). If the WA valorization were already defined, CLCA could be an interesting option to modelized the different WA treatment scenarios. However, as so many hypotheses are required to carry out an CLCA, the results would only provide recommendation for each specific scenario, whereas the aim of this paper was to be more generic. The discussion part has been enriched with such comments.

  • On another matter, the aspects of biogenics and dynamics of carbon emissions on global warming calculations has only been minorly covered. Please discuss this more thoroughly at least.

Answer: Indeed it was only minorly covered as the CML-IA methodology cannot support such calculations and we also considered the global assumption that the CO2 captured and stored during the biomass growth is entirely released during the end-of-life of this biomass.

  • Similar literature on the same topic is still not well considered, e.g. such as the work of Gaudreault et al. (2020) and Tosti et al. (Tosti et al., 2020). Please compare with these works and showcase the novelty compared to these.

Answer: those both articles are dealing with different waste treatments of WA only, assuming that it is a waste generated from the heat production, so no impact can be attributed to produce these WA. The boundaries of their studies are starting where we stop ours (at the BPP gate). They are this not pertinent to compare our results are we are not on the same perimeter. However, they illustrate greatly that WA is more and more studied to be used as a resource within industrial processes, so its status of waste or coproduct should be examined and that is exactly the point of our work.

Mind typos such as “global worming”

Answer: these have been corrected and a thoroughly proof-readding was carried out.

Reviewer 2 Report

The authors implemented my improvement suggestions and I consider the revised manuscript fit for publishing.

Author Response

Thank you for your comment

Reviewer 3 Report

congratulations

Author Response

Thank you for your comment

Round 3

Reviewer 1 Report

The authors have addressed a comment in an acceptable way